# Identification of Candidate mRNA Isoforms for Prostate Cancer-Risk SNPs Utilizing Iso-eQTL and sQTL Methods

**DOI:** 10.3390/ijms232012406

**Published:** 2022-10-17

**Authors:** Afshin Moradi, Harsh Sharma, Ravi Datta Sharma, Achala Fernando, Roberto A. Barrero, Jyotsna Batra

**Affiliations:** 1Centre for Genomics and Personalised Health, Queensland University of Technology, Brisbane 4059, Australia; 2Translational Research Institute, Queensland University of Technology, Brisbane 4102, Australia; 3Faculty of Health, School of Biomedical Sciences, Queensland University of Technology, Brisbane 4059, Australia; 4Amity Institute of Integrative Sciences and Health, Amity University Haryana, Gurugram 122413, India; 5eResearch, Research Infrastructure, Academic Division, Queensland University of Technology, Brisbane 4000, Australia

**Keywords:** alternative spicing, prostate cancer, Iso-eQTL, sQTL

## Abstract

Single nucleotide polymorphisms (SNPs) impacting the alternative splicing (AS) process (sQTLs) or isoform expression (iso-eQTL) are implicated as important cancer regulatory elements. To find the sQTL and iso-eQTL, we retrieved prostate cancer (PrCa) tissue RNA-seq and genotype data originating from 385 PrCa European patients from The Cancer Genome Atlas. We conducted RNA-seq analysis with isoform-based and splice event-based approaches. The MatrixEQTL was used to identify PrCa-associated sQTLs and iso-eQTLs. The overlap between sQTL and iso-eQTL with GWAS loci and those that are differentially expressed between cancer and normal tissue were identified. The cis-acting associations (FDR < 0.05) for PrCa-risk SNPs identified 42, 123, and 90 PrCa-associated cassette exons, intron retention, and mRNA isoforms belonging to 25, 95, and 83 genes, respectively; while assessment of trans-acting association (FDR < 0.05) yielded 59, 65, and 196 PrCa-associated cassette exons, intron retention and mRNA isoforms belonging to 35, 55, and 181 genes, respectively. The results suggest that functional PrCa-associated SNPs can play a role in PrCa genesis by making an important contribution to the dysregulation of AS and, consequently, impacting the expression of the mRNA isoforms.

## 1. Introduction

Alternative splicing (AS) is a conserved biological process that allows the generation of several different mRNA isoforms from one gene locus [1,2,3]. This process occurs in close relation to transcription in eukaryotes [4]. AS is subject to tissue-, cell type-, or condition-specific regulation [5]. Abnormal splicing patterns have been detected in cancers, and these patterns are involved in carcinogenesis and metastasis [6,7]. Researchers have discovered several cancer-specific mRNA isoforms [8]. One well-known example is the androgen receptor (AR) isoforms in prostate cancer (PrCa) [9]. Alternative splicing events in the AR locus yield different types of mRNA isoforms, including those harbouring cryptic exons downstream of the coding sequence [9], lacking exons due to exon skipping [10], and changes in the open reading frame (ORF), causing the translation of shorter proteins that are not androgen dependent [11,12,13]. Furthermore, the trimming of a C-terminal ligand-binding domain in some AR variants can make them resistant to traditional androgen deprivation therapies [14,15].

Most of the trait-associated genetic variants such as single nucleotide polymorphisms (SNPs) have uncharacterised functions [16]. A fraction of genetic variations may function as a regulator of gene-splicing events. The chromosomal loci harbouring these SNPs are also known as splicing quantitative trait loci (sQTLs) [17]. Moreover, the chromosomal loci harbouring SNPs that impact mRNA isoform expression are known as isoform-expression quantitative trait loci (iso-eQTLs) [18]. Detection of sQTL and iso-eQTLs, particularly in cancerous cells, can be used in the service of genomic medicine [19,20]. For example, the fine mapping of GWAS associations to isoform loci can lead to the identification of isoforms that correspond to a GWAS disease. Another usage of iso-eQTL study is highlighting likely causal isoforms among differentially expressed isoforms [21].

In this study, we aim to investigate mRNA isoforms that are associated with PrCa. In the first step, after RNA-seq analysis (ensemble), we performed iso-eQTL analysis and integrated its results into the GWAS catalogue for prioritising the causal mRNA isoforms that correspond to PrCa. Then, we found those associated isoforms that are differentially expressed between cancer and normal tissue to find causal ones in PrCa, on the assumption that only the co-expressed isoforms that also harbour a disease association are causally involved [21]. In the second step, we first conducted sQTL analysis for exon skipping and intron retention; and then, we found prostate-associated splicing using GWAS loci. Next, considering the position of exon and intron, we predicted the structure of new PrCa-associated mRNA isoforms. These outcomes have the highest probability of highlighting the isoforms most likely to contribute to PrCa progression.

## 2. Results

### 2.1. Identifying Iso-eQTLs and sQTL

The generation and processing of the genotype and RNA-seq data are detailed in the method section and Figure 1. We included genotyped SNPs in 385 unrelated individuals of European ancestry [22]. To identify the sQTL and iso-eQTL, cis- analysis was performed for all types of cassette exon events, retention events, and isoform expression. Considering an FDR 0.05 cut-off, we identified 446,793 and 167,294 significant cis-sQTL for intron retention and cassette exon, respectively.

Regarding iso-eQTL, we identified 245,636 cis-iso-eQTLs. Detailed tables of this analysis are represented in Appendix A. By analysing cis-acting associations and focusing on PrCa-risk SNPs, we discovered 42, 123, and 90 PrCa-associated cassette exons, intron retention, and mRNA isoforms belonging to 25, 95, and 83 genes, respectively (Appendix A). Many variants affect splicing patterns by altering the sequence of splice sites and many more modify the binding sites of RNA-binding proteins.

Considering FDR < 0.05, we identified 86,010 and 72,853 significant trans-sQTLs for intron retention and cassette exon, respectively. We identified significant 239,497 trans-iso-eQTLs, considering FDR < 0.05 (Appendix A); we identified 59, 65, and 196 PrCa-associated cassette exons, intron retention, and mRNA isoforms belonging to 35, 55, and 181 genes, respectively (Appendix A). Then, we scanned the sQTL and iso-QTL list and extracted the list of isoforms and splicing that are differentially expressed between PrCa cancer and normal tissue (Appendix A). Every sQTL and iso-eQTL table displays the SNP ID of the sQTL and iso-eQTL analysis, SNP alleles, SNP genomic position (hg19), minor allele frequency (MAF) of SNP, event ID (or isoform ensemble ID), β-value (effect size estimate), state (test statistic (t-statistic of T-test)), *p*-value, FDR (False discovery rate estimated with the Benjamini–Hochberg procedure), minor allele frequency (MAF) of the SNP identified in sQTL and iso-eQTL, average call of the SNPs, and the Rsq of the imputation.

The list of prostates iso-eQTL SNPs and sQTL SNPs can be used to identify candidate variants causally associated with PrCa. The top five most significant cis-cassette-exons-sQTL are EX76463, EX260111, EX291121, EX561966, and EX273766, which are in *cingulin like 1 (CGNL1)*, *zinc finger protein 506 (ZNF506)*, *X-ray radiation resistance associated 1 (XRRA1)*, *ENSG00000262879,* and *protein kinase C delta (PRKCD)* genes, respectively. The *CGNL1* promotes angiogenesis via Rac1 activation, which leads to stabilisation and further elongation of newly formed vascular tubules [23]. The low *ZNF506* mRNA levels in breast and lung cancers correlated with lower survival probability and worse prognosis [24]. The *X-ray radiation resistance associated 1 (XRRA1)* has been suggested to regulate the response of human tumour and normal cells to X-radiation (XR) [25]. *Protein kinase C-delta* inactivation inhibits the proliferation and survival of cancer stem cells in culture and in vivo [26]. The top five significant cis-intron-retention-sQTL are INT123326, INT98520, INT145829, INT79370, and INT144093 located in the *adenosine A2b receptor (ADORA2B)*; *cytochrome c oxidase assembly factor 1 (COA1)*; *protein phosphatase 1 regulatory inhibitor subunit 11 (PPP1R11)*; *dynein 2 intermediate chain 1 (DYNC2I1)*; and *immunoglobulin mu DNA binding protein 2 (IGHMBP2)*. *Adenosine receptor 2B* activity promotes autonomous growth, migration, as well as vascularization of head and neck squamous cell carcinoma cells [27]. *Cytochrome C oxidase assembly factor 1 Homolog* predicts poor prognosis and promotes cell proliferation in colorectal cancer by regulating PI3K/AKT signalling [28]. *IGHMBP2* promotes cell migration and invasion in esophageal squamous carcinoma by downregulating E-cadherin [29]. The top five significant cis-iso-eQTL are ENST00000460851.6 *(EIF2A*-203), ENST00000437043.8 (*ERAP2*-202), ENST00000620995.5 (*TYW1B*-203), ENST00000495526.5 (*PPIE*-214), and ENST00000397119.8 (*DYNC1l2*-202). Between them, ENST00000495526.5 was also discovered in normal prostate tissue sQTL [5]. The list of sQTL and iso-eQTL SNPs among cancer-associated loci shows that some of these SNPs could contribute to the risk of cancer by affecting alternative splicing and isoform expression. The lists of GWAS-sQTL and GWAS-iso-eQTL are available in Appendix A.

The association *p*-values of multiple SNPs on autosomal chromosomes for all cis analyses are shown in Manhattan plots (Figure 2a, Figure 3a and Figure 4a). A volcano plot depicts the differentially expressed isoform and splicing in cis analysis (Figure 2b, Figure 3b and Figure 4b). We showed the position of SNPs in cis analysis (FDR 0.05) via the density plots (Figure 2c, Figure 3c and Figure 4c). Distribution plots show the number of associated SNPs based on distance from the isoform and splicing events. Interestingly, the associated SNPs with isoform expression and splicing are mostly located within 200 Kb and by going further, the number of associated SNPs decreases significantly. The association *p*-values of multiple SNPs for all trans analysis are shown in Manhattan plots (Figure 2e, Figure 3e and Figure 4e). A volcano plot (Figure 2f, Figure 3f, and Figure 4f) depicts the differentially expressed isoform and splicing of trans analysis.

### 2.2. Isoform-Structure Prediction

An isoform structure matrix is a binary matrix, wherein 1 indicates the inclusion of a meta feature in the given mRNA isoform and 0 indicates its exclusion. We predicted mRNA isoform structure for 25 events. We found mRNA isoforms from differentially expressed cassette exon and differentially expressed intron retention. For example, three mRNA isoforms related to the Lemur tyrosine kinase 2 (LMTK2) gene were found. The first and second isoforms are differentially expressed in tumor and normal tissue. The only difference between these two isoforms is an intron retained region (INT133250) between exon 7 and 8 in rs9649213, which is in the PrCa-associated GWAS loci (Supplementary File S1).

## 3. Discussion

AS can occur in many ways; i.e., exons can be extended or skipped, or introns can be retained [30]. AS can be determined either by differential mRNA isoform expression or by using meta-feature level expression. Isoform-based methods estimate mRNA isoform structure and rely on known mRNA isoform annotations. Meta-feature methods can estimate the position of splicing events [31]. Some SNPs can modulate diseases in human individuals through their impact on AS [32,33,34]. This study identifies casual mRNA isoforms in PrCa by integrating sQTL, iso-eQTL, GWAS SNPs, and differentially expressed mRNA isoforms in tumour and normal tissues. The findings in this study suggest that many of the PrCa-associated germline variants can impact AS and change the expression of mRNA isoforms and, consequently, might be functionally impacted in PrCa pathogenesis. For example, kallikrein-related peptidases (KLKs), serine peptidases, are critical regulators of the tumor microenvironment. *KLK* expression can be regulated by alternative splicing [35]. One important outcome of this study is the discovery of genetic variation that can regulate AS in KLK3 that is used as a biomarker to indicate the presence of PrCa in patients; we found that the expression of ENST00000360617.7 (KLK3-202), ENST00000593997.5 (KLK-204), ENST00000596333.1 (KLK3-209), and ENST00000598145.1 (KLK3-212) KLK3 mRNA isoforms are regulated by SNPs in chromosome 19 loci in a cis manner. Between them, the expression of KLK3-202, KLK-204, and KLK3-209 is driven by PrCa risk-associated SNPs (GWAS catalog). These SNPs might be one example of a causal SNP that could indirectly promote cancer progression by altering KLK3 isoform expression, which can impact the tumor microenvironment. Another example is the LMTK2 gene, which is a modulator of androgen receptor activity. We predicted the structures of new isoforms in this locus that are differentially expressed in tumor and normal tissue.

Further exploration of the mechanisms of the action of the identified iso-eQTL and sQTLs could lead us to discover a mechanism of dysregulated AS in PrCa (37). Our results provide a new insight into the genetic architecture of PrCa that can be functional through splicing. Integrating the PrCa risk loci and sQTL information prioritises likely causal variants in PrCa that impact AS (51–53). In particular, incorporating the sQTL and iso-eQTLs as functional germline variants into the clinic may enhance the speed of translational research. Furthermore, some of these sQTL and iso-eQTL SNPs have the potential to contribute to cancer risks. Some of these mRNA isoforms that are differentially expressed may serve as biomarkers. In conclusion, two different pipelines were applied to discover casual mRNA isoforms in PrCa. Further studies are required to investigate and confirm the role of those mRNA isoforms in PrCa progression.

## 4. Materials and Methods

### 4.1. Data Collection, Genotype Data pre-Processing and Imputation

Genotype data (Affymetrix 6.0 arrays) and RNA-seq (bam format) originated from normal tissue and PrCa tumour tissues were downloaded from The Cancer Genome Atlas (TCGA) (https://tcga-data.nci.nih.gov/tcga/ (accessed on 20 Febuary 2020)). The SNP quality control was performed using genotype data. Individuals with more than 2% missing genotypes were removed. SNPs with a call rate of less than 98% and those not in Hardy–Weinberg equilibrium (HWE) (threshold *p*-value < 10^−8^) were excluded. SNPs with a minor allele frequency (MAF) of less than 5% were excluded from further analysis. Since a high level of heterozygosity shows low sample quality, whereas low levels of heterozygosity may be due to inbreeding [36], we filtered out the samples with high heterozygosity (mean heterozygosity ±3 standard deviation (s.d.)) using PLINK 1.9 and R [36]. All the samples were checked for gender discrepancies between their recorded sex in the dataset and their genetically determined sex. Genetic studies with individuals of admixed ancestries can confound genetic association studies. The principal component analysis (PC) is performed, and individuals who are outliers of the European population were removed (PC1 and 2 ± 6 s.d. from the average). Prior to the imputation, indels or non-biallelic variants, in addition to the ambiguous SNPs that did not match the reference panel or were duplicates, were excluded using quality control offered by the Michigan server [37]. The genotype data were imputed using the 1000 Genomes Project phase 3 reference panel. Genotypes were imputed using the Minimac3 software (v1.0.5) provided by the Michigan Imputation Server. Analyses were limited to the SNPs with MAF > 0.5% and a squared correlation coefficient between imputed allele dosages and masked genotypes *r*^2^  >  0.5 [37].

### 4.2. Isoform Expression Analysis

RNA-seq Bam files were sorted and converted to Fastq format using samtools and bcftools, respectively. For RNA-Seq Analysis, Snakemake Workflow (RASflow) was used [38]. For this analysis, HISAT2 was selected for alignment to the transcriptome, feature-count for quantification, and DEseq2 for normalisation, batch effect removal, as well as differential expression analysis.

### 4.3. Splicing Event Data

We used FASE (https://github.com/harshsharma-cb/FASE (accessed on 1 January 2021)) where ExonPointer and IntronPointer algorithms are inbuilt [39]. Both methods utilised information from meta features (flanking junctions and skipping junctions), along with the event in question (exon/intron). For example, the ExonPointer algorithm is based on the principle that if an exon is expressed in one condition, its flanking junction (s) should be expressed, and its skipping junction (s) should not be expressed; and vice-versa in another condition. Similarly, the IntronPointer algorithm detects intron-retention events. If an intron is expressed, its skipping junction (s) should not be expressed and vice-versa [39]. Both methods detect differential alternative splicing events using summarised meta-feature counts normalised with the help of the TMM method inbuilt in the edgeR software package (version: 3.15) that finds associations and ranks outputs using summarised scores equivalent to *p*-values.

### 4.4. Iso-eQTL and sQTL Analysis

The MatrixEQTL software (R package; version: 2.3) that uses a linear regression model was utilised for miR-eQTL analysis. Then, PC1 to PC10 were used as variables to adjust for the population stratification. The *cis* and *trans* associations were calculated for the PrCa primary tissue sample. In *cis-* analysis, SNPs in proximity to the target splicing event (or mRNA isoform) on the same chromosome were chosen, while for *trans-* analysis, SNPs on different chromosomes or the same chromosome further than 1 MB from their target splicing event (or mRNA isoforms) were selected. MatrixEQTL performs multiple testing corrections using the Benjamini–Hochberg method [22] to estimate FDR.

### 4.5. GWAS Related Iso-eQTL and sQTL

The list of GWAS SNPs for the available data on 16 cancers—i.e., bladder, breast, cervical, colon, endometrial, gastric, head and neck, kidney, lung, oral, ovarian, pancreatic, prostate, skin, stomach, and thyroid—were downloaded from the NHGRI website [40]. GWAS linkage disequilibrium (LD) regions (up to 1 MB distance) were calculated with plink 1.9 [41] using the 1000 Genomes phase 3 reference panel. By using this list of SNPs within 1 MB regions around GWAS SNPs, we extracted iso-eQTL and sQTL SNPs that are in LD with a cancer-associated SNP identified in the GWAS catalogue at *r*^2^ > 0.5.

### 4.6. Isoform-Structure Prediction

To perform isoform structure prediction, the meta features that were expressed in at least one mRNA isoform were included. The FASE pipeline was used for isoform prediction (https://github.com/harshsharma-cb/FASE (accessed on 20 April 2022)) using differential alternative splicing events obtained through the ExonPointer and IntronPointer algorithms. It predicts unique mRNA isoforms based on differential alternative splicing events (cassette exon and intron retention) using RNA-Seq data. It finds meta features (exons, introns, flanking junctions, and skipping junctions) associated with the event and applies graph theory to find the corresponding mRNA isoform structures. We predicted mRNA isoform structures for 25 events.

## Figures and Tables

**Figure 1 ijms-23-12406-f001:**
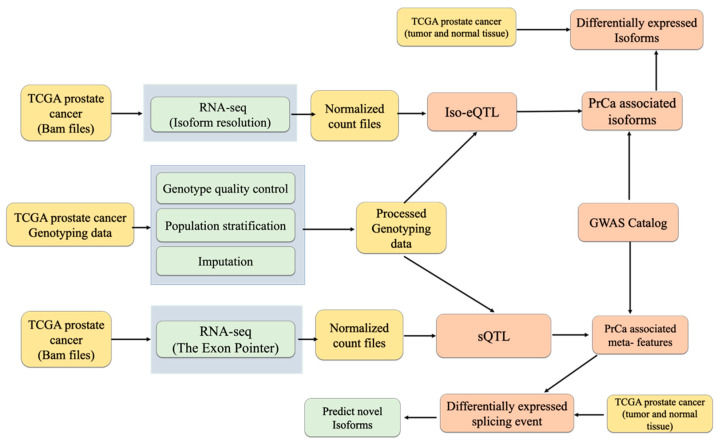
The sQTL and iso-eQTL analysis pipeline. The yellow boxes show the types of the data, the green boxes show the methods of analysis, and the orange boxes show the outcomes of this analysis.

**Figure 2 ijms-23-12406-f002:**
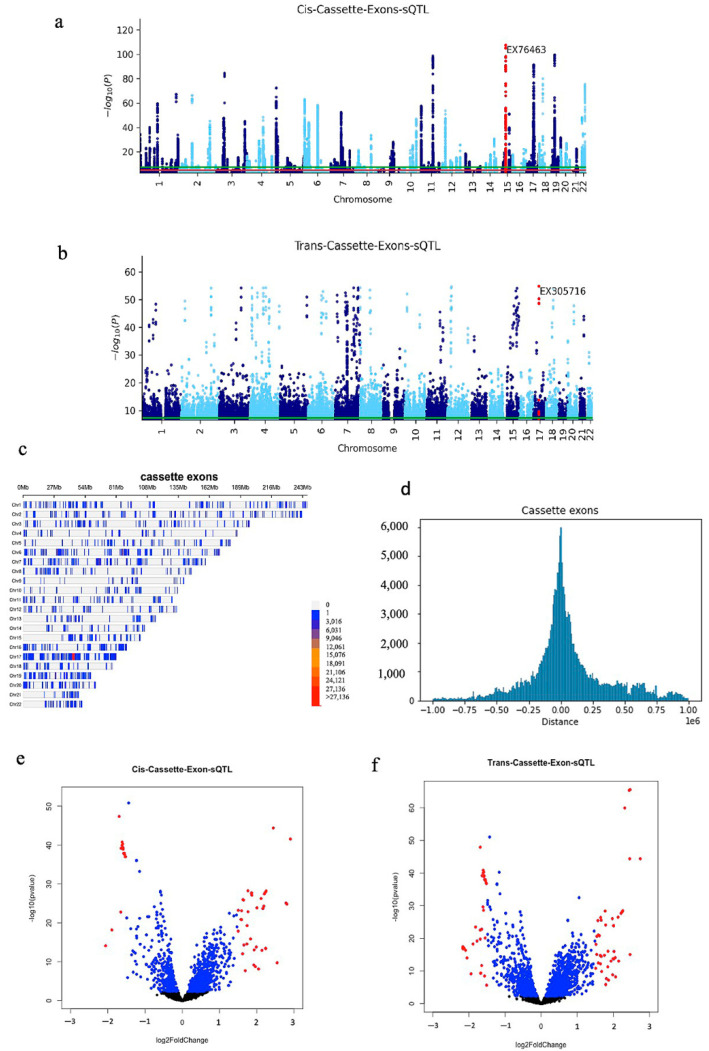
(**a**) Manhattan plots of cis-cassette-exons-sQTL; the most significant event is labelled. (**b**) Manhattan plots of trans-cassette-exons-sQTL. (**c**) The density plot shows the position of all cis-cassette-exons-sQTL (FDR 0.05) in the genome. (**d**) Distribution plots of the distance between the position of the SNP and the position of the splicing events in the cis-cassette-exons-sQTL. (**e**) Volcano plot of differentially expressed cassette-exons that are in the cis-sQTL list; FC > 1.5 is shown in red. (**f**) Volcano plot of cassette-exons in the trans-sQTL list; FC > 1.5 is highlighted in red.

**Figure 3 ijms-23-12406-f003:**
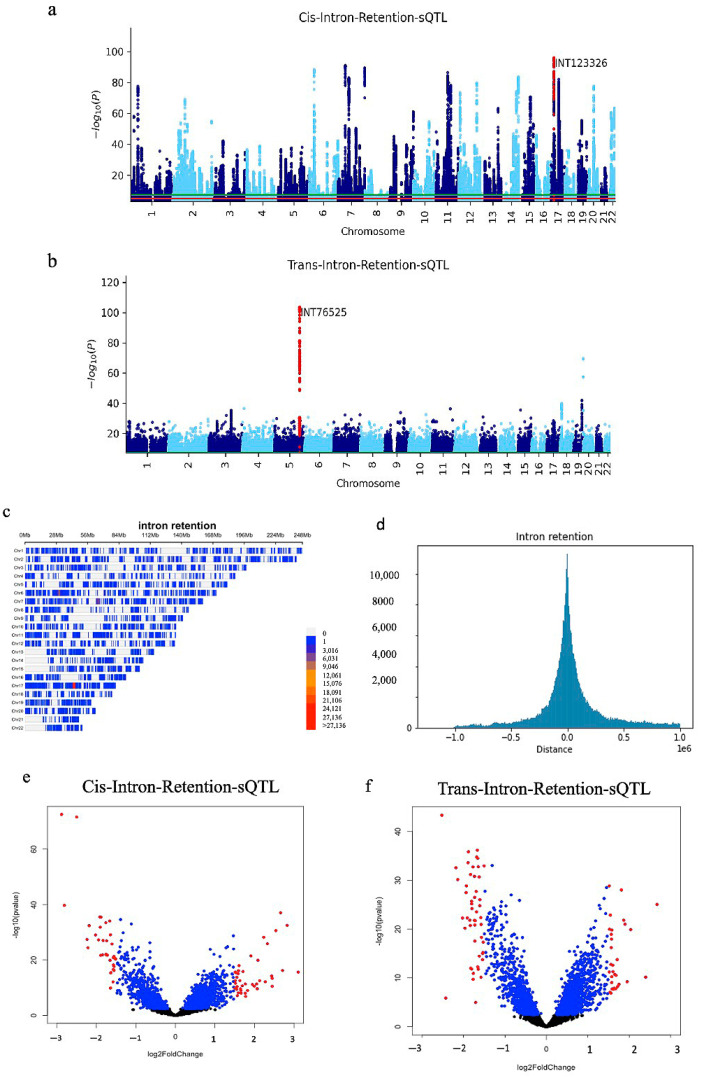
(**a**) Manhattan plot of cis-intron-retention-sQTL; the most significant event is labelled. (**b**) Manhattan plot of trans-intron-retention-sQTL. (**c**) The density plot shows the position of all cis-intron-retention-sQTL (FDR 0.05) in the genome. (**d**) Distribution plots of the distance between the position of the SNP and the position of the splicing events in the cis-intron-retention-sQTL. (**e**) Volcano plot of intron-retention in the trans-sQTL list; FC > 1.5 is highlighted in red. (**f**) A volcano plot of differentially expressed intron-retention in the cis-sQTL list, with FC > 1.5 highlighted in red.

**Figure 4 ijms-23-12406-f004:**
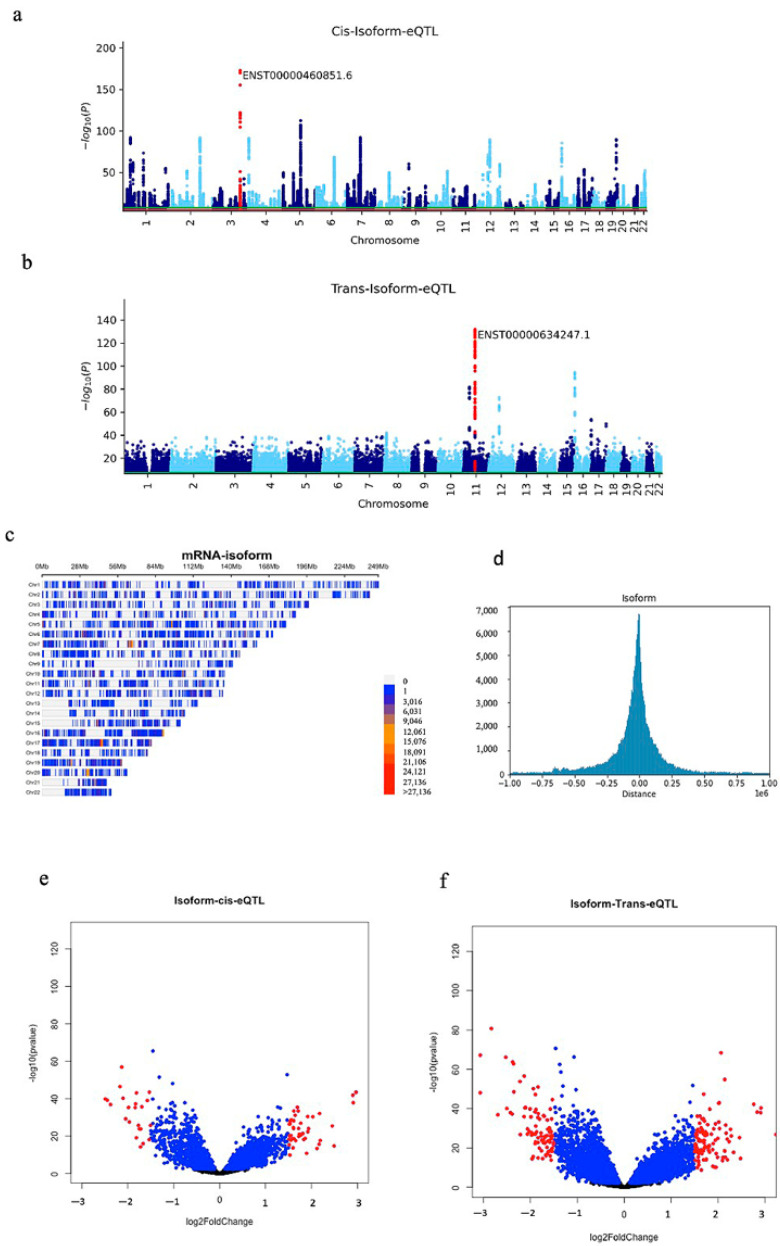
(**a**) Manhattan plot of iso-eQTL; the most significant event is labelled. (**b**) Manhattan plot of trans-iso-eQTL. (**c**) The density plot shows the position of all cis-iso-eQTL (FDR 0.05) in the genome. (**d**) Distance distribution plots between SNP and isoform positions in cis-iso-eQTL. (**e**) Volcano plot of differentially expressed isoforms that are in the cis-iso-eQTL list. FC > 1.5 is shown in red. (**f**) Volcano plot of isoform expression that is in the trans-eQTL list; FC > 1.5 is shown in red.

## Data Availability

The results of this study are in whole based upon data generated by the TCGA Research Network: https://www.cancer.gov/tcga (accessed on 20 February 2020).

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
