# Peer review of "Identification of Candidate mRNA Isoforms for Prostate Cancer-Risk SNPs Utilizing Iso-eQTL and sQTL Methods"

_ijms, 2022, doi:10.3390/ijms232012406_

Round 1

Reviewer 1 Report

The authors of the manuscript took up an important topic, i.e. the identification of a specific mRNA isoform that would be an indicator of the risk of prostate cancer. Due to the large number of cases of this disease, this topic should be intensively researched.

The research was designed in an appropriate way, using appropriately selected techniques. The description of the results is quite synthetic but understandable. The reviewer failed to explain in the manuscript why such a high FDR (5%) was used, while usually in large-scale analyzes the FDR should not exceed 1%.

Figure 2 should also be corrected, which is now horizontal and should be vertical.

Author Response

Response: We believe 5 % is suitable cut off for FDR value. 1 % is really restrict for this analysis. Bellow we added couple of examples that used 5%.

https://www.ncbi.nlm.nih.gov/pmc/articles/PMC6943077/

https://www.frontiersin.org/articles/10.3389/fgene.2020.00486/full

https://academic.oup.com/nar/article/46/D1/D971/4210944?login=false

Figure 2 should also be corrected, which is now horizontal and should be vertical.

Response: It is corrected

Reviewer 2 Report

I would like to suggest improving Fig. 2, 3, and 4 in regard to resolution formatting and increasing the font size of the text so it is readable.
Please revise the Introduction to avoid any Plagiarism.

Author Response

1- I would like to suggest improving Fig. 2, 3, and 4 in regard to resolution formatting and increasing the font size of the text so it is readable.

Response: In agreement with the Review’s comment. We have improved the resolution and font size of Figures 2, 3 and 4.

2- Please revise the Introduction to avoid any Plagiarism.

Response: We acknowledge the Reviewer comment and in the revised manuscript, we revised a paraph in the introduction section (lines 37-42)

Reviewer 3 Report

The authors used TCGA data to identify PrCa-associated sQTLs and iso-eQTLs. 

  1. No figures or tables support the results described in section “2.2. Isoform-structure prediction” (line 271-280). I couldn’t find Supplementary Table 19 in the supplementary materials. 

  2. Figure 2 is not shown properly. 

  3. Figure caption doesn’t match the panels in Figure2-4.

  4. Where is the RNA-seq data of normal tissue from? The Materials and Methods session only mentioned that the data of PrCa tumour tissues were from TCGA but no information for the normal tissue. 

  5. If the data of normal tissue was not from TCGA, how did the authors take care of the batch effects?

Author Response

1-No figures or tables support the results described in section “2.2. Isoform-structure prediction” (line 271-280). I couldn’t find Supplementary Table 19 in the supplementary materials. 

Response: We acknowledge the Reviewer comment. The support evidence of the results described in section “2.2. Isoform-structure prediction” are presented in Supplementary files 1. This has been corrected in the revised manuscript (lines 275-284).

2- Figure 2 is not shown properly. 

Response: In agreement with the Reviewer, we have modified Figure 2 resolution and increased the fonts sizes for visibility.

3- Figure caption doesn’t match the panels in Figure2-4.

Response: Following the Reviewer comment, captions for Figures 2-4 have been corrected.

4-Where is the RNA-seq data of normal tissue from? The Materials and Methods session only mentioned that the data of PrCa tumour tissues were from TCGA but no information for the normal tissue. If the data of normal tissue was not from TCGA, how did the authors take care of the batch effects?

Response:

We acknowledge the Reviewers’ feedback. Normal tissues were also downloaded from TCGA. This have been highlighted in the revised text (line 326).

Also, for ‘4.2 Isoform expression analysis’ we added reference to batch effect removal using DESeq2 when comparing TCGA normal vs tumour tissues (line 351). Furthermore, Figure 1 has been also updated to clarify the use of both normal and tumour tissue for differential expression analysis and identifying differential expression events.

Round 2

Reviewer 3 Report

The authors have addressed my concerns and I don't have further comments.